# Cultural contexts differentially shape parents' loneliness and wellbeing during the empty nest period
Andree Hartanto [1] ✉, Lester Sim[1], Davelle Lee [1], Nadyanna M. Majeed[2] & Jose C. Yong [3]

The coming decades will see a substantial increase in the population of older adults, accompanied by significant demographic and family structure changes worldwide. As a result, the empty nest period— the postparental phase in parents' lives when their children have left home and they are no longer engaged in childrearing—is becoming an increasingly common experience in Western and Asian cultures. The current theoretical review examines the psychological consequences of the empty nest period on loneliness and well-being across cultures, emphasizing the impact of cultural factors on these experiences. By synthesizing research from Western and Asian contexts, we explore two primary theoretical mechanisms—role loss and role strain relief—that shape the postparental phase's psychological outcomes. Our review reveals that while some parents experience reduced well-being due to role loss, others benefit from role strain relief and increased social engagement. We highlight how cultural differences in familial roles, gender roles, social expectations regarding nest-leaving, and social participation patterns moderate these mechanisms. We propose a comprehensive cultural framework, along with a discussion of culturally sensitive interventions to enhance the well-being of empty nesters globally.

In the coming decades, the world will witness an unprecedented increase in the number of older adults. The number of people aged 65 years or older is projected to double within the next 20–30 years[1], with global life expectancy expected to rise from 73.3 years in 2024 to 77.4 by 2054, coupled with a significant reduction in fertility rates[2]. Alongside this demographic shift, significant changes in family structures have emerged. Women's child-bearing years have been compressed since the 1970s[3], with both a later onset of childbearing and a shorter span of years in which childbearing occurs, thereby extending the period parents live after their last child leaves home. Additionally, industrialization has led to a significant decrease in family size and an increase in geographic distance between nuclear family members from the 1970s to 2010[4,5]. The evolving family structure, coupled with the demographic shifts, have led to the rise of the empty nest period—defined as the stage when parents no longer have dependent children in residence[6]—as a common life transition across the world. Although the empty nest is a relatively modern phenomenon that emerged in the 20th century, it has now become a normative event anticipated by older adult parents across many cultures[7–9].

Recent demographic data further substantiate the widespread prevalence of the empty nest phenomenon. In Western societies, the phenomenon of the empty nest has been well-documented over the past several decades. For instance, the United States saw a 10.6% increase in the number of empty-nester households from 2010 to 2020, with 37% of householders aged 45–64 and 44% of those aged 65 and above living without children[10]. Similarly, in Canada, the proportion of couples living without children grew by 5.5% from 2016 to 2021, making up 25.7% of all households[11]. These trends reflect a growing prevalence of empty nesters in Western societies, driven by factors such as decreased family sizes and increased geographic distance between nuclear family members[4,5].

In Asian societies, the empty nest phenomenon is also becoming more prevalent. Traditionally, many young adults remained in the parental home, creating multigenerational households due to cultural norms such as filial piety; even when they left, they did so at a later age[12,13]. However, modernization and urbanization have significantly altered family structures in Asia. Young adults are increasingly migrating to urban areas for employment, leaving their parents in rural regions. Data from recent studies conducted in China and South Korea, which show that urban migration of young adults has led to a significant increase in the number of elderly parents living alone, especially in rural areas[14,15]. In China, for example, there are ~150 million empty nesters, accounting for more than half of the elderly

¹School of Social Sciences, Singapore Management University, Singapore, Singapore. ²Department of Psychology, National University of Singapore, Singapore, Singapore. ³Department of Psychology, Northumbria University, Newcastle-upon-Tyne, United Kingdom. ✉e-mail: andreeh@smu.edu.sg

population[14]. This number is projected to rise significantly, with empty nesters expected to make up 90% of older adult households by 2030[16–18]. Similar patterns are observed in other Asian countries, where declining fertility rates are compounding the trend. Major economies such as Japan, South Korea, India, Malaysia, and Singapore have all reported plummeting fertility rates, contributing to a growing population of empty nesters[19–24].

Given the growing population of empty nesters worldwide, it is crucial to understand the impact of this postparental phase on loneliness and well-being of older adults, as well as its underlying mechanisms and boundary conditions. As with other life transitions, the empty nest period is associated with changes in well-being, which can be both positive and negative[25–28]. Research has indeed shown that parents' emotional experiences during this phase are mixed[8]. For example, various studies have shown that the post-parental period is associated with reduced well-being, with postparenthood perceived as stressful[27,29,30], socially isolating[26,31], and associated with loneliness and depressive symptoms[32–36]. Conversely, other studies indicate that parents may experience enhanced well-being or reduced ill-being during this period, particularly when they maintain regular contact with their children[37–40]. Some studies have also found null effects, suggesting no evidence for any significant changes in well-being during the postparental phase[28,41,42]. The inconsistency in these findings suggests the existence of multiple opposing mechanisms at play in determining the well-being of empty nesters[43].

Over the last 50 years, a vast body of research has been accumulated on the empty nest period in Western society to shed light on these mechanisms (see Bouchard[43] for a review). More recently, researchers in Asia have also turned their focus to this issue, likely in response to the region's rapidly aging population[28,32,44–47]. The current paper aims to synthesize the research on the psychological consequences of the empty nest period on older adults' loneliness and well-being across cultures and critically review its underlying mechanisms and boundary conditions through a cultural lens.

## Theoretical mechanisms

The empty nest period is a significant milestone that typically involves transitioning from at least two decades of dedicated parenting to the abrupt withdrawal from such activities. As with any other important life transition, this can substantially influence the psychological experience and well-being of those involved. Several theories have been advanced to understand this phenomenon.

### Role loss

According to the role loss model[43], the transition to the empty nest period can hurt the well-being of older adults due to the loss of domestic or work roles that are associated with or contribute to shaping one's identity[48]. Identity theories posit that having an identity is fundamental to one's sense of self and serves as an important reference point to ascertain one's purpose in life[49,50]. The roles individuals play contribute to a sense of identity, and these roles come with expectations and norms that guide behavior. Parenthood is often viewed as a central role that provides a sense of accomplishment and efficacy for both parents, especially so for mothers[43]. When the last child leaves home, parents face the abrupt end of their parental duties and sudden changes in their daily routines, disrupting their sense of normalcy and stability and leading to uncertainty about their new reality[43,44]. This loss can also create a void in parents' identities, as they may not have other roles to fill the gap[40]. The resulting uncertainty, emptiness, and lack of purpose can contribute to decreased well-being and lead to significant emotional distress, including symptoms of depression and anxiety[51–56]. Moreover, the altered parent-child relationship can be particularly distressing. As children become independent adults, parents may feel unneeded and undervalued, which is emotionally challenging for those who derive much of their self-worth from caregiving[57,58]. Additionally, the loss of the parental role can strain marital relationships by bringing underlying issues to the forefront, requiring couples to renegotiate their dynamics[8,59,60].

### Role strain relief

On the other hand, the role strain relief model suggests that the empty nest period can improve well-being by relieving parents of the difficulties experienced while fulfilling multiple role obligations during parenting[43]. Most parents continue to hold a job while raising their children, doubling their daily demands and creating significant pressure on themselves[61–65]. Moreover, many parents set high standards for their involvement in their children's lives[66]. This increased involvement means that parents are constantly juggling the demands of their careers and family life, leading to greater work-family conflict and heightened stress levels[67–71]. Furthermore, the presence of children, including adult children living at homes, exposes parents to numerous daily stressors, such as managing household chores, cooking meals, providing ongoing emotional support, and financial responsibilities[72–74]. These responsibilities can be overwhelming and leave parents with little time for themselves, contributing to chronic stress and fatigue[75,76].

When children leave home, parents often experience significant relief from these childcare stressors, providing them with more time and energy to focus on their own needs and interests[40]. This newfound freedom allows parents to engage in activities they enjoy, pursue personal goals, and strengthen social connections that may have been neglected. Studies have shown that this reduction in role strain can lead to improvements in overall well-being, including better mental health, and increased life satisfaction[77]. Additionally, the empty nest period can provide an opportunity for parents to reconnect with their partners. Without the constant demands of parenting, couples can spend more quality time together, engage in shared activities, and communicate more effectively[78,79].

### Social engagement

Beyond the direct effects of role loss and role strain relief on well-being, these experiences during the empty nest period also differentially impact older adults by altering their levels of social engagement. Research consistently shows that social engagement is strongly linked to well-being in older adults[80–82]. Positive social interactions, especially face-to-face interactions, can buffer against loneliness and depression, increase life satisfaction, and enhance overall well-being[83–86]. Older adults involved in volunteer work, clubs, or community events often experience these benefits[87,88]. Importantly, interactions with familiar people, especially one's children, significantly influence well-being more than formal or solitary activities[89–92].

Role loss when children leave the nest may result in a substantial reduction in parents' social interactions with their children, leading to feelings of loneliness and decreased well-being[28,93]. This reduction in daily social contact can exacerbate the emotional void left by the departure of children, making it more difficult for parents to adjust to their new reality[94]. The lack of regular contact with children means parents must seek alternative social networks and activities to fill the gap left by their children's departure[95]. In contrast, role strain relief can foster increased social engagement. When the daily demands of parenting are lifted, older adults often find themselves with more time, energy, and resources to pursue new social opportunities[77,96]. This can include volunteering, joining clubs, participating in community events, or reconnecting with old friends. Engaging in these new roles and activities can help older adults build new social networks, fostering a sense of community and belonging[97,98].

Understanding the co-occurring and opposing mechanisms of role loss and role strain relief on social engagement and well-being provides insight into the mixed findings on the psychological consequences of the empty nest period[43]. These mechanisms illustrate how some parents might experience increased well-being through newfound freedoms and social engagements, while others might struggle with isolation and a lack of purpose. Next, we propose that the moderating role of culture may further explain individual and intergroup differences in the psychological impact of the empty nest period.

## The role of culture

Culture plays an important role in shaping people's experiences during the empty nest period, influencing role loss, role strain relief, and subsequent social engagement. Research has established cultural differences in familial roles[99,100], gender roles[101,102], social expectancies related to nest-leaving[103,104], and patterns of social participation[105,106]. These cultural differences can potentially explain variations in the psychological consequences of the empty nest period. The following sections illustrate the influence of several factors on empty nesters' psychological experience through a comparison of Eastern and Western cultures.

### Familial role

Cultural norms have been shown to affect how individuals perceive their familial roles[99,100,107,108], which can influence their psychological adjustment to the empty nest phase. Cultural differences in familial roles can be understood through the lens of individualism–collectivism, a key cultural dimension that captures the importance of relationships with members of one's ingroup (e.g., family, extended kin) in one's conceptualization of the self[109,110]. Western societies are typically characterized as individualistic, where the self is viewed as an independent, bounded entity. In contrast, Asian societies are characterized as collectivistic, where the self is seen as part of a larger social unit, deeply intertwined with others. Consequently, the family unit holds greater significance for collectivists compared to individualists, leading to stronger family ties in collectivistic cultures[111].

In collectivistic cultures, ascribed roles, duties, and obligations often take precedence over personal goals[109,112,113]. Familial roles are clearly and rigidly defined within a hierarchical (and often, but not always, patriarchal) family structure[99,100]. In many Asian societies, including those in East Asia, South Asia, and Southeast Asia, parents have a strong sense of interdependence with their children, influencing the degree of role strain experienced in parenthood[114–117]. Collectivists tend to view work as an integral part of their familial responsibilities, considering it a means to support and strengthen the family unit rather than a competing obligation, which could explain studies that have found less work-family conflict in these cultures due to the alignment of work and family goals[118–122]. However, in some collectivistic cultures, parents may experience greater parenting stress due to authoritarian parenting values and the emphasis on fostering children's self-regulation and academic success[123,124].

For example, in Confucian cultures, the teachings dictate that children demonstrate filial piety, showing care and respect for their parents throughout their lives[100,125,126]. This expectation extends well beyond the child's marriage and formation of their own family. Consequently, the postparental period may relieve stress for collectivistic parents[124] while allowing them to gain a sense of interdependence and well-being given that their children are obligated to maintain strong familial bonds and care for parents in return[127,128]. Additionally, Asians' acceptance of age-graded roles and expectations may make them more accepting of the loss of the child-rearing role in middle or old age compared to Westerners[129]. Thus, while the parent-child relationship is crucial for collectivists, role loss may not result in as significant a decrease in well-being as it does for individualists, for whom the parental role is central[130].

In individualistic cultures, the parental role is often one of many personal roles an individual may hold[109]. Western parents tend to emphasize personal autonomy and the development of their children's independence, leading to a different kind of parental satisfaction and stress[131,132]. When children leave the nest, individualistic parents may experience role loss as a natural progression toward their children's self-sufficiency, which they value highly. However, this can also lead to a sense of redundancy and a need to rediscover their personal identity outside the parental role. The emphasis on personal goals and self-fulfillment may mean that individualists are more likely to seek new opportunities for self-development and social engagement postparenthood, potentially mitigating the negative effects of role loss.

Western societies, with their greater focus on individual achievement and self-actualization, often encourage parents to pursue personal and professional goals alongside parenting[109,110,112]. This can lead to significant role strain as parents juggle multiple responsibilities, but it also means that the relief from parenting duties during the empty nest period can provide substantial psychological benefits[41,43]. The transition can offer a chance to re-engage with personal interests and social activities that were previously limited by parenting responsibilities, thereby enhancing quality of life and well-being.

### Gender roles

Traditional Western societies dictate that the mother-child relationship is categorically different from the father-child relationship. The feminine gender role prescribes that women should focus substantially on domestic affairs and devote their time to nurturing their children and family[133]. Men, on the other hand, are often seen as the leaders of the household, providing financially and making important decisions on behalf of the family. This socialization leads men and women to perceive their parental roles differently. Stay-at-home mothers, unlike their working husbands, experience parenthood as a central part of their self-identities[101]. Consequently, role loss tends to have a greater impact on women than men in traditional societies, which may explain why various studies have found that the empty nest period is associated with poorer well-being for women but not for men[27,33]. However, some men also report feeling a sense of loss during the empty nest period, often due to a perceived lack of opportunities for new accomplishments or experiences[134]. With their children gone, men may feel that their role as providers is diminished, and they may struggle to find new avenues for meaningful engagement and achievement. On the other hand, women's greater burden of care means that working women experience greater work-family conflict compared to working men[135,136]. As a result, working women are likely to experience greater relief from role strain after their children have left the nest compared to men.

Interestingly, it is proposed that Western women may cope better with this significant role loss compared to men facing a similar life transition[133,137]. While men do not show significant changes in well-being in response to the empty nest, they are vulnerable to the loss of their work role. Since being a breadwinner is a key facet of the masculine gender role[47,133], retirement can be experienced as a great loss. Men in retirement experience more depressive symptoms compared to those who engage in paid or volunteer work, while for women, there was no significant difference in levels of depression across work statuses[137]. Sugihara and colleagues[137] speculated that this was because women had frequent informal social activities beyond their work spheres, which protected them from the loss of social engagement.

In Asian settings, traditional gender roles also increase the impact of role loss and role strain for women in postparenthood. For example, Japanese women do more than five times as much unpaid work at home as men[137], including childrearing. Similar to Western women, Asian women typically experience some degree of role loss when their children leave home[8]. However, they may also experience positive feelings towards their new age-related role as empty nesters, compared to Westerners who may feel negatively about stepping back from parenting[129]. Additionally, Asian parents tend to view nest-leaving as an age-appropriate transition for daughters, provided that the child is leaving for socially acceptable reasons, and are thus more accepting of it[8].

On top of having gendered parental roles, Asians' gender roles extend to children's caregiving responsibilities, which may reduce the likelihood of experiencing the empty nest period altogether. The eldest son of the family is expected to live with his parents and be responsible for their care and well-being[138–141]. This caregiving burden ultimately falls on the daughter-in-law, who is expected to be the primary caregiver for her husband's elderly parents[142–145]. Because Asian women are expected to care for their parents-in-law, their caregiving role may extend beyond their children's years of dependence, continuing to experience role strain postparenthood. Traditional Asian gender roles promote intergenerational co-residence, in which a significant proportion of aging parents are unlikely ever to have to face an empty nest. Indeed, the rate of intergenerational co-residence is much higher in Asia (e.g., 68.7% in China[146]) than it is in the West (e.g., 14% in the United States[147]). Furthermore, modern Asian parents now expect both daughters and sons to care for them in their old age[148].

## Social expectancies on nest-leaving

Cultures vary significantly in the social expectancies surrounding the empty nest period, such as the timing and meaning of a child's departure. These social expectancies directly impact parents' perceptions of their own parental and postparental roles. In Western societies, it is generally expected that children will leave home upon reaching adulthood, typically between the ages of 18 and 25, although the exact timing varies across countries[7,149,150]. Western parents tend to expect their children to establish themselves outside the home for socially acceptable reasons, such as expressing independence, pursuing education or a career, or getting married. However, variations exist depending on socio-economic factors, such as housing affordability and labor market conditions, which have led to delayed nest-leaving in many contexts[150,151]. Parents who anticipate this transition but experience delays may report difficulty coping[8], a distress possibly stemming from parents' role strain and their desire to be released from the parental role. When expectations are met, parents experience pride and satisfaction in seeing their children leave the nest[43]. For some, this act of nest-leaving is perceived as a sign of the child's maturity; for others, their children's success reflects their own success as parents[58]. An older adult's ego integrity can be bolstered by the achievements of their offspring[152], which can buffer against the impact of role loss.

Conversely, parents who did not expect their children to leave already tend to experience more anxiety during the empty nest period[58,153]. Parents typically expect children to leave home after developing emotional and financial independence and are prone to worry if they do not believe their child is prepared to live alone[58,154]. Some Western parents, particularly women, continue to expect a high degree of involvement in their children's lives even after they have moved out. For example, some mothers report feeling obliged to continue caring for their children by doing household chores such as cooking and laundry[58]. These women may be attempting to prevent the loss of their parental role while simultaneously prolonging their experience of role strain.

In contrast, collectivistic societies have different expectations regarding when and why children should leave the nest. For instance, Chinese and Korean adult children in urban areas are often not expected to move out of the parental home even after marriage due to the high cost of housing[155–157]. In exchange for staying, children provide their parents with an allowance, creating mutual dependence that meets both parties' expectations and avoids role loss or role strain. While Westerners approve of a variety of reasons for nest-leaving, including establishing independence, Asians tend to perceive marriage as the most legitimate reason for moving out and show significantly less approval for nest-leaving before marriage[158]. Nest-leaving for reasons other than what is socially sanctioned, such as seeking independence, can be distressing to Asian parents, who may perceive it as a threat to the family unit[103,159]. This type of nest-leaving could challenge the older adult's parental role, indirectly producing a sense of role loss and resulting in anxiety and stress[104]. However, this is not true for all Asian countries. In Thailand, for example, parents show lower levels of depression when children live further from rather than closer to the parental home[160] because their successful migration confers a sense of pride and higher social status for the parents, thus enhancing ego integrity[152] and mitigating the impact of role loss. This example highlights the cultural specificity of norms and expectations surrounding nest-leaving.

Moreover, it is common in Asian cultures to expect grandparents to continue being heavily involved in caregiving for grandchildren[161,162]. As Asians' standards of childcare have risen alongside their education levels[66], grandparents may be held to higher expectations by their children or children-in-law, leading to more stressful intergenerational interactions[163] and greater grandparental role strain.

## Social participation patterns

The transition into the empty nest period engenders a dramatic change in the parent-child relationship among both Asians and Westerners[29]. Parents not only relinquish their caregiving role but also lose the opportunity for daily social interactions with their children. Consequently, one key challenge elderly people face is loneliness, stemming from a lack of social interactions[164,165]. While social participation and support benefit empty nesters[166–169], the means through which older adults achieve social integration and obtain social support may differ across cultures[106,170]. According to socioemotional selectivity theory[171], older adults in Western societies tend to prioritize close social relationships over more distant acquaintances as they age, increasing emotional closeness and thereby enhancing their well-being. For Westerners, having the autonomy to choose who to socially engage with is a significant predictor of well-being[172]. Consequently, interactions with friends and close confidantes are strongly associated with a reduction in loneliness, while interactions with family members, which are often seen as obligatory, are not as strongly linked to reduced loneliness. Among Americans, the size of one's non-familial social network tends to increase with age[173]. Conversely, East Asians with their higher levels of interdependence are more likely to maintain a diverse social network, particularly as they age[174]. As age increases, East Asians tend to retain or increase the number of close contacts and are less likely to reduce the number of peripheral social partners. Only Asians with low interdependence resemble Westerners by maintaining a small group of close others[175].

Despite these differences, empty nesters from Asian cultures are more prone to social isolation than those in Western cultures[172]. In Asia, age-graded social norms often encourage older adults to reduce their social network involvement[129]. A recent study in China found that empty nesters were more likely to be excluded from social relationships, experience feelings of exclusion, and participate less in social activities compared to non-empty nesters[26]. This social isolation correlates with higher reports of depression among empty nesters, and those living without a spouse were the most likely to suffer from loneliness and social isolation[176,177]. Comparing empty nesters to older adults with residential children in China revealed no significant differences in subjective well-being; instead, living alone predicted lower well-being[177]. Similarly, elderly Japanese individuals living in single-person households had significantly greater social isolation and loneliness scores than those in multi-person households[178].

## Diversity in family structures

Cultures vary significantly in societal norms that create distinct family arrangements, many of which are more widely accepted in the West than in Asia (e.g., divorced, single-parent, stepparent families). Recent shifts in Western societal norms have also profoundly transformed family dynamics, giving rise to newer family structures that implicate the empty nest (e.g., boomerang children, crowded nests, etc.)[75,179]. While some of these diverse family structures are beginning to emerge in certain Asian societies, traditional cultural norms continue to define what constitutes a typical family unit; deviations from more traditional structures (i.e., marriage) remain taboo. These contrasting attitudes towards family structures affect how patterns of early departure and potential returns may unfold, shaping the empty nest experience in different ways.

Divorced, single-parent, and stepparent families represent some of the diverse family arrangements seen in contemporary societies. In Western societies, young adults (particularly daughters) from divorced, single-parent, and stepparent families are more likely to leave their parents' homes earlier than their counterparts from traditional two-biological parent families[13,180,181]. This early departure accelerates the empty nest experience for parents. Some reasons for this trend include the parental home becoming a less comfortable and divestment of limited parental resources shared amongst a larger number of children[13,182]. Notably for divorced and single-parent families, early nest leaving can lead to a loss in practical and financial support from adult children, and contribute negatively to a heightened sense of loneliness and role displacement for older adult parents, as their absence of a partner can intensify the emotional vacuum left by children. In contrast, research on divorced, single-parent, as well as step-families remains less common in Asia due to strong cultural norms that emphasize traditional family units[183,184]. The limited scholarship suggests that while divorce rates and non-traditional family structures are increasing in some Asian countries, these trends have not yet significantly altered the

patterns of adult children leaving the family home. In fact, children in Asia from similar family structures may stay longer due to cultural expectations surrounding family duty and support[12,13]. For adult children that do leave the nest, there is often a sense of loss, which can create further stress and hardship for the parents left behind, both emotionally and practically, as older parents have to navigate aging, psychological, and potential health challenges alone[185].

Tangentially, a significant pattern that gained research traction is the rise of the "boomerang generation"— adult children, often those without adequate emotional and financial stability, who return to live with their older parents after initially leaving home due to factors such as unemployment, and high housing costs[153,186,187]. This pattern of intergenerational coresidence can re-introduce conflict and strain to the parent-child relationship, while simultaneously, conferring new opportunities for quality interactions to develop between child and adult parents[13,180]. A related issue arises when some older siblings return to the family home while younger adult siblings are still living there[188]. Older parents in the "crowded nests" might face elevated levels of chronic stress, fatigue, and compromised well-being as a result of managing ongoing childrearing demands, with the addition of household responsibilities and chores[75]. Tension between parents and their adult children increases, and heightened role strain might dampen well-being. Research on boomerang children has gained more attention in recent years in Asia, including Taiwan, China, and Thailand[189,190]. As in the West, many adult children return home primarily due to economic reasons. Yet in Asia, it is not uncommon for the boomerang phenomenon to occur. In fact, adult children are likely to return to the parental home after having their own children to save resources and money on childcare due to a shortage of public childcare facilities and the high cost of private childcare in their respective countries[190]. In all, evolving social norms are slowly reshaping family structures and arrangements in Asia that could eventually lead to changes in the timing and reasons for adult children leaving home. Although the extent of this shift remains less pronounced compared to Western societies, they have significant consequences on the empty nest and adult parents' well-being.

## A culturally sensitive framework for empty nesters

As the empty nest period becomes an increasingly common experience in both Western and Asian societies, there is a growing need to understand the psychological well-being of empty nesters and the conditions under which this phase can be a positive experience for older adults. Studies have shown that parents' emotional experiences during the empty nest period are mixed[8]. Some research indicates that the postparental phase can be detrimental to parents' well-being, with postparenthood perceived as a stressful and depressive experience[27,30,36] and associated with increased loneliness and social isolation[26,31,33,34]. Conversely, other studies suggest that parents may experience enhanced well-being or reduced ill-being during this period[38–40]. Given these mixed findings, there is a need for an integrative framework of the postparental phase to reconcile the existing heterogeneity in the literature.

Having reviewed the existing literature from a cultural perspective, we propose a comprehensive framework to understand the psychological mechanisms underlying the empty nest period and identify boundary conditions that account for the observed variability in psychological experiences. Our theoretical framework posits that cultural factors play a significant role in shaping the psychological experience of the postparental phase during the empty nest period. Specifically, cultural differences in familial roles, gender roles, social expectancies on nest-leaving, and social participation patterns result in different outcomes in role loss and role strain relief. These differences subsequently influence social engagement, thereby impacting parents' loneliness and well-being. Figure 1 provides a summary of the theoretical framework of the empty nest period through a cultural lens.

A crucial implication of this theoretical framework is the need for culturally sensitive interventions to support the well-being of empty nesters. Given the significant role of culture in shaping the experience of the empty nest period, interventions that are effective in one cultural setting may not necessarily be effective in another. For instance, happiness interventions such as practicing gratitude and performing acts of kindness were less effective in improving South Koreans' well-being as compared to that of Americans[191]. Instead, in collectivistic cultures, interventions that facilitate

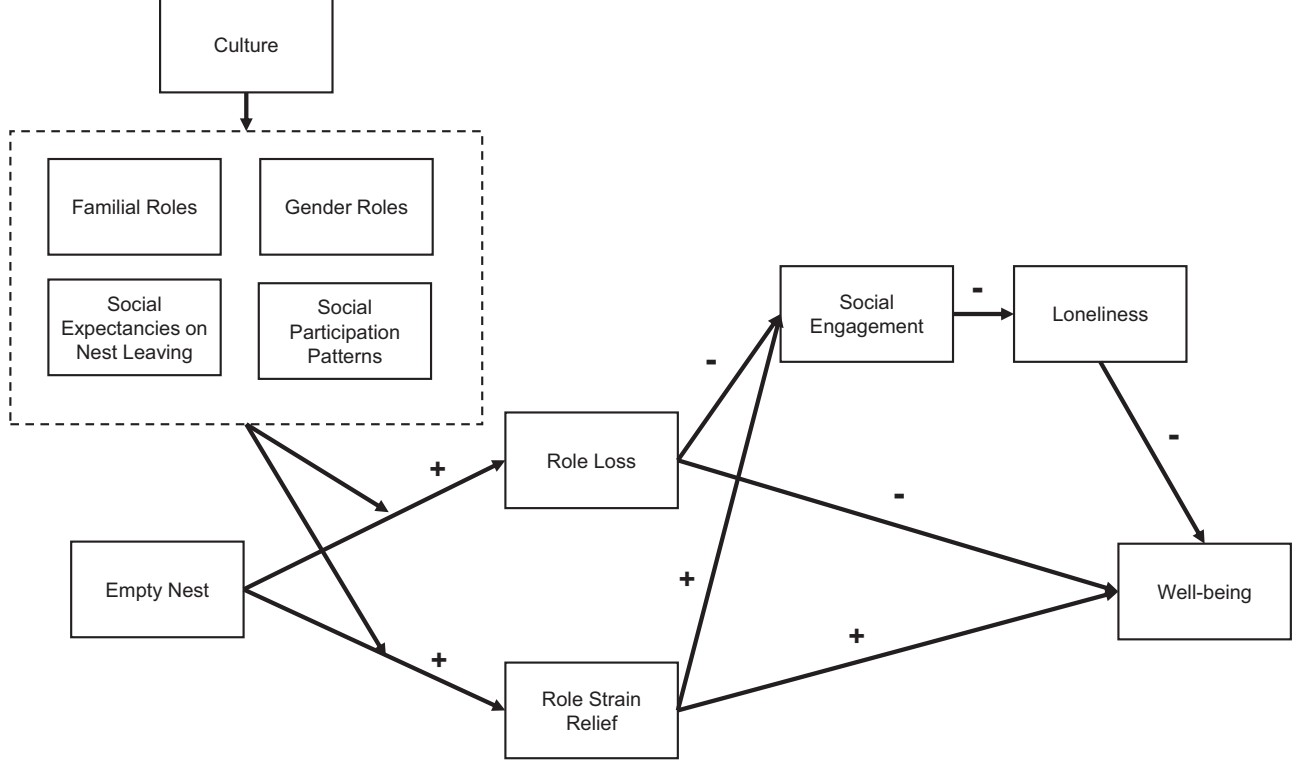

**Fig. 1 | Cultural framework of the empty nest period.**

more frequent family interactions, such as Singapore's incentive policies for intergenerational housing or tax relief for caregiving[24], may be more beneficial given their higher levels of familism[100,126,192].

In contrast, the stronger emphasis on individual autonomy might drive Western societies to focus more on helping parents find new personal goals and social engagements outside the family unit. Programs such as Switzerland's KISS, which matches older adults with people in need and allows them to earn credits for volunteering, have been shown to enhance regular social contact for participants[193]. Similarly, Finland's Circle of Friends program, an occupational therapy-based intervention, has successfully increased feelings of connectedness among older adults[194]. These initiatives highlight how different cultural contexts necessitate tailored approaches to effectively support empty nesters.

Relatedly, when discussing family structures, including the empty nest phenomenon, it is essential to recognize the profound diversity within Asian societies. While many discussions (and empirical research) on socially accepted timelines for departures from the parental homes in Asia often anchor on Confucian values, especially filial piety, which are prevalent in East Asian countries like China, Japan, and South Korea, these values do not uniformly define family norms across the entire continent[195,196]. For instance, in Southeast Asia, extended family support and co-residence are common, but the motivations for these living arrangements and expectations for adult children are rooted more on cultural and religious beliefs (e.g., Islam, Buddhism, and Hinduism), than in East Asia[197]. Similarly, South Asian countries such as India, Pakistan, and Bangladesh maintain strong traditions of multigenerational households, where there is often an expectation that the nest never empties: at least one child—often a son and his wife — remains at home to establish an extended household to care for aging parents[197]; violations to these cultural expectations can result in stress and anxiety for parents[104]. These patterns are reinforced by religious and cultural norms that vary widely from East Asian Confucian ideals.

Even within Asian countries, there are significant differences in family practices between urban and rural areas, as well as between socioeconomic classes. Urbanization and modernization are reshaping family dynamics in countries like China and India, but this shift is not uniform, even within those countries. Rural areas are more likely to continue to hold onto traditional family structures, whereas urban families may adopt more "Westernized" living arrangements, such as nuclear families or young adults leaving home earlier[146]. Given this complexity, it is essential for researchers to avoid overgeneralizing when examining the empty nest phenomenon in Asia given the rich tapestry of family forms, norms, and transitions that differ not only between countries but within them. As such, we suggest researchers should engage in more nuanced and localized research to cast spotlight on the uniqueness for how empty nest plays out within different Asian cultural contexts. Culturally sensitive interventions also need to address these variations, and fit contemporary cultural norms, the state of economic development, and urbanization to better address the empty nest experiences of older parents and their adult children across the diverse societies of Asia.

Given the increasing use of information and communication technology (ICT) for video chatting, gaming, and social networking[198–200], there is also a pressing need for more cross-cultural research to assess the efficacy of ICT-related interventions to reduce social isolation and loneliness among empty nesters and older adults living alone[201,202]. Social networking sites can provide a means to stay in frequent contact with non-residential children as well as to maintain some involvement in their lives by viewing or commenting on their posts[94,203,204]. Additionally, innovative solutions such as virtual reality avatars and companion robots are being developed to support older adults[205,206]. However, acceptance and effectiveness of these technologies can vary across cultures. Asian empty nesters, who may value interdependent relationships, might be less open to using new technologies such as robots for companionship[207], due to their preference for close, personal connections which these technologies cannot yet easily replicate. At the same time, much of the research pertaining to ICT-related interventions targeting social participation and loneliness has been conducted in Western

countries[208]. Thus, more cross-cultural research is necessary to determine whether ICT can enhance Asian empty nesters' well-being and to guide the design of more culturally sensitive ICT interventions.

## Conclusion

In conclusion, the empty nest period is a complex transition influenced by a multitude of factors, including cultural norms, gender roles, social expectations, and individual family dynamics. By adopting a culturally sensitive framework, we can better understand the diverse experiences of empty nesters and develop targeted interventions to support their psychological well-being. As the global population ages, addressing the needs of empty nesters will become increasingly important in promoting healthy aging and ensuring a high quality of life for older adults.

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

## Acknowledgements

This research was supported by grants awarded to Andree Hartanto by Singapore Management University through research grants from the Ministry of Education Academy Research Fund Tier 1 (22-SOSS-SMU-041). The funders had no role in the preparation of the manuscript or the decision to publish.

## Author contributions

A.H. and D.L. conceptualized the paper and drafted the manuscript. A.H., L.S., D.L., N.M.M., and J.C.Y. contributed to the critical revision of the manuscript. All authors contributed to the article and approved the submitted version.

## Competing interests

The authors declare no competing interests.
