## [Transparent Peer Review file · Communications Psychology]

Cultural Contexts Differentially Shape Parents' Loneliness and Wellbeing During the Empty Nest Period

Corresponding Author: Dr Andree Hartanto

Version 0:

Decision Letter:

Dear Dr Hartanto,

Thank you for your patience during the peer-review process. Your manuscript titled "A Theoretical Framework on Loneliness and Well-Being During the Empty Nest Period Across Cultures" has now been seen by 2 reviewers, and I include their comments at the end of this message.

The reviewers find the piece in principle of interest, but point to some significant shortcomings. We are interested in the possibility of publishing your study in *Communications Psychology*, but would like to assess a revised manuscript before we make a final decision on publication.

In particular, we ask for the following core issues to be addressed:

First, to be suitable as a review-type paper in *Communications Psychology*, the work must cover the most recent relevant literature and provide a balanced and up-to-date overview of work in the field.

Second, and related to the issue of relevant literature, the revised version of the manuscript would need to achieve greater clarity and compelling justifications for the focus on specific age groups, and national, cultural or ethnic groups.

We therefore invite you to revise and resubmit your manuscript, along with a point-by-point response to the reviewers. Please highlight all changes in the manuscript text file.

Please use the following link to submit your

- revised manuscript,
- point-by-point response to the referees' comments,
- cover letter (as a separate document),

Link Redacted

We hope to receive your revised paper within 8 weeks; please let us know if you aren't able to submit it within this time so that we can discuss how best to proceed. If we don't hear from you, and the revision process takes significantly longer, we may close your file.

Best regards,

Marika Schiffer

Marika Schiffer, PhD
Chief Editor
Communications Psychology

REVIEWER EXPERTISE:

Reviewer #1 Loneliness, social isolation, ageing
Reviewer #2 Loneliness, ageing, global studies

REVIEWER REPORTS:

Reviewer #1 (Remarks to the Author):

The paper is a synthesis of studies exploring the experience of empty nesters and loneliness. The paper explores the different mechanisms under which parents whose children have left home may experience loneliness of provide opportunities for enhanced social interaction, and how this is shaped by culture and gender. The paper is well written, and presents a theoretical framework to support the central argument that cultural factors have a central role in shaping the post parental experience. As a synthesis of existing literature, there is no novelty in the findings, having said that it will be of interest to others in the community, and to social gerontologists studying family dynamics, or loneliness. Some of the references are over a decade old with some over 20 years old and could be updated with more recent publications if available. So some minor revision of references would be beneficial.

Reviewer #2 (Remarks to the Author):

I always enjoy reading papers that address big conceptual issues. In this paper, the authors feature a family life course stage that came into the family life course lexicon in western countries in the mid 20th century. I agreed to review the paper because I was intrigued about how someone might now be using the idea of 'empty nest', at a point in history in which there is such immense diversity in family structure that the author rightly identifies in the first section of the paper.

Part of the argument for focusing on the empty nest is that it has become more common in both Western and Asian societies. It would be helpful if this claim could be substantiated. The empty nest period is described as "the postparental phase in parents' lives when their children have left home and they are no longer engaged in childrearing". We might be able to determine how many people of a particular age cohort have ever had children, although it could be difficult given that the age range implied in this review could include anyone from age 40 up to very late life. I wonder too if it matters whether 'their children' have always lived with them in the same household, or whether they have come together in a blended family, or when the nest is counted as empty. Barbara Mitchell, whose work which is cited in the paper, talks about children who leave and then return and about how in the context of Canada and a housing crisis, children are staying home longer.

The authors argue that loneliness is associated with lack of daily contact with children in some cultures. Does it matter whether they are living alone, whether they depend on their children for financial or other support, whether they live in rural or urban areas or whether a parent moves in with a child in late life? The challenge I see is that the variation across all of these demographic and other social factors is immense. I'd love to see some critical thinking about the usefulness of the concept of empty nest given this diversity.

It would be a good idea for the authors to be a bit more vigilant about statements that rely on literature that sometimes is decades old. One example is a statement by Rick Settersten from 1998 that "In the West, the common consensus is that children, regardless of gender, should leave home between the ages of 18 to 25 (Settersten, 1998). That statement was made 25 years ago by a family theorist in the U.S. I counted almost 50 references in the paper that are 20+ years old. There's nothing wrong with citing classic studies. I'd like to see them identified as such.

In conclusion, I think the premise of the paper is very bold. It needs a bit of tweaking and perhaps a narrower focus to provide the insightful conclusions that we'd like to see in a theoretical paper.

EDITORIAL POLICIES

We ask that you ensure your manuscript complies with our editorial policies and reporting requirements.

To that end, we require revised manuscripts to be accompanied by two completed items: a reporting summary that collects

information on study design and procedure, and an editorial policy checklist that verifies compliance with all required editorial policies.

- <https://www.nature.com/documents/nr-reporting-summary.zip>>Nature Research Reporting Summary
- <https://www.nature.com/documents/nr-editorial-policy-checklist.pdf>>Editorial Policy Checklist

All points on the policy checklist must be addressed. Your revised manuscript can only be sent back to the referees if these checklists are completed and uploaded with the revision.

Notes: If you have submitted a Stage 1 Registered Report, Review, Primer, Comment, or Perspective you do not need to submit these forms. If you have already submitted these forms, you may disregard this request.

Communications Psychology is committed to improving transparency in authorship. As part of our efforts in this direction, we are now requesting that all authors identified as 'corresponding author' create and link their Open Researcher and Contributor Identifier (ORCID) with their account on the Manuscript Tracking System prior to acceptance. ORCID helps the scientific community achieve unambiguous attribution of all scholarly contributions. You can create and link your ORCID from the home page of the Manuscript Tracking System by clicking on 'Modify my Springer Nature account' and following the instructions in the link below. Please also inform all co-authors that they can add their ORCIDs to their accounts and that they must do so prior to acceptance.
<https://www.springernature.com/gp/researchers/orcid/orcid-for-nature-research>

Version 1:

Decision Letter:

** Please ensure you delete the link to your author homepage in this e-mail if you wish to forward it to your co-authors **

Dear Dr Hartanto,

Your Article titled "A Theoretical Framework on Loneliness and Well-Being During the Empty Nest Period Across Cultures" has now been editorially vetted. I am delighted to say that we are happy, in principle, to publish it as a Perspective in Communications Psychology.

We will not send your revised paper for further review if, in the editors' judgement, all remaining issues are addressed. If the revised paper is in Communications Psychology format, in accessible style and of appropriate length, we shall accept it for publication immediately. I have attached an edited version of your manuscript and an Editorial Request Table and ask you to attend to each comment in detail.

EDITORIAL REQUESTS:

* Please review the changes in the attached copy of your manuscript, which has been edited for style, and address the comments and queries I have added. If using Word, please use the 'track changes' feature to make the process of accepting your manuscript more efficient.

* Communications Psychology uses a transparent peer review system. On author request, confidential information and data can be removed from the published reviewer reports and rebuttal letters prior to publication. If you are concerned about the release of confidential data, please let us know specifically what information you would like to have removed. Please note

that we cannot incorporate redactions for any other reasons.

*If you have not done so already, please alert me to any related manuscripts from your group that are under consideration or in press at other journals, or are being written up for submission to other journals (see www.nature.com/authors/editorial_policies/duplicate.html for details).

* References

References appear as superscript Arabic numerals, in order of mention. The reference list mentions references in the numerical order in which they are mentioned in the main text. If a reference is cited more than once, the same number is used throughout the text and the reference receives a single entry in the reference list.

We ask that you select the most significant 5–10% of references in your list for highlighting, and add a single sentence in bold after each of these references to describe the main result and its significance.

Only papers that have been published or accepted by a named publication should be in the reference list (preprints and citations of datasets are also permitted). Unpublished/Submitted research should not be included in the reference list; it should only be mentioned briefly and parenthetically in the main text. Note that no major arguments should rely on unpublished research.

Published conference abstracts and URLs for web sites should be cited parenthetically in the text, not in the reference list.

* Competing interests

Please include a "Competing interests" statement after the References. Note that we ask authors to declare both financial and non-financial competing interests. For more details, see <https://www.nature.com/authors/policies/competing.html>. If you have no financial or non-financial competing interests, please state so: "The authors declare no competing interests."

SUBMISSION INFORMATION:

* If you wish, you may also submit a visually arresting image, together with a concise legend, for consideration as a 'Hero Image' on our homepage. The file should be 1400x400 pixels and should be uploaded as 'Related Manuscript File'. In addition to our home page, we may also use this image (with credit) in other journal-specific promotional material.

In order to accept your paper, we require the following:

* A cover letter describing your response to our editorial requests.

* The final version of your text as a Word or TeX/LaTeX file, with any tables prepared using the Table menu in Word or the table environment in TeX/LaTeX and using the 'track changes' feature in Word.

* Production-quality versions of all figures, supplied as separate files. Photographic images should be 300 dpi in RGB format (.jpg, TIFF or native Photoshop format) and any labels/scale bars included in a separate layer from the image. Line art, graphs and schemes should be vector format (.ai, .eps, .pdf); Adobe Illustrator files are preferred and will minimize production time. Any chemical structures or schemes contained within figures should additionally be supplied as separate Chemdraw (.cdx) files.

Please note that your paper cannot be sent for typesetting to our production team until we have received this information; **therefore, please ensure that you have this ready when submitting the final version of your manuscript.**

ORCID

Communications Psychology is committed to improving transparency in authorship. As part of our efforts in this direction, we are now requesting that all authors identified as 'corresponding author' create and link their Open Researcher and Contributor Identifier (ORCID) with their account on the Manuscript Tracking System (MTS) prior to acceptance. ORCID helps the scientific community achieve unambiguous attribution of all scholarly contributions. For more information please

visit <http://www.springernature.com/orcid>

For all corresponding authors listed on the manuscript, please follow the instructions in the link below to link your ORCID to your account on our MTS before submitting the final version of the manuscript. If you do not yet have an ORCID you will be able to create one in minutes.

IMPORTANT: All authors identified as 'corresponding author' on the manuscript must follow these instructions. Non-corresponding authors do not have to link their ORCIDs but are encouraged to do so. Please note that it will not be possible to add/modify ORCIDs at proof. Thus, if they wish to have their ORCID added to the paper they must also follow the above procedure prior to acceptance.

To support ORCID's aims, we only allow a single ORCID identifier to be attached to one account. If you have any issues attaching an ORCID identifier to your MTS account, please contact the [Platform Support Helpdesk](http://platformsupport.nature.com/).

Link Redacted

We hope to hear from you within two weeks; please let us know if the process may take longer.

Best regards,

Marike

Marike Schiffer, PhD
Chief Editor
Communications Psychology

Ms. Ref. No.: COMMSPSYCHOL-24-0363-T

Title: A Theoretical Framework on Loneliness and Well-Being During the Empty Nest
Period Across Cultures

Dear Dr. Marike Schiffer:

We thank the Reviewers for their thorough and incisive reviews, which have significantly strengthened the revised paper. We strove to carefully address both Reviewers' concerns and suggestions and hope that this revision is much stronger. Our point-by-point responses are listed and detailed below, along with the Reviewers' comments.

Thank you once again for your consideration of our paper.

Reviewer #1

The paper is a synthesis of studies exploring the experience of empty nesters and loneliness. The paper explores the different mechanisms under which parents whose children have left home may experience loneliness of provide opportunities for enhanced social interaction, and how this is shaped by culture and gender. The paper is well written, and presents a theoretical framework to support the central argument that cultural factors have a central role in shaping the post parental experience.

1. *As a synthesis of existing literature, there is no novelty in the findings, having said that it will be of interest to others in the community, and to social gerontologists studying family dynamics, or loneliness. Some of the references are over a decade old with some over 20 years old and could be updated with more recent publications if*

available. So, some minor revision of references would be beneficial.

Response:

We thank the Reviewer for the positive feedback and the constructive comments. We appreciate the Reviewer's recognition that the paper will be of interest to social gerontologists and scholars studying family dynamics and loneliness, particularly due to its focus on cultural and gender factors shaping the postparental experience. We also appreciate the Reviewer's suggestion to revise our citations to reflect more recent publications. In response, we have carefully reviewed and updated the manuscript by replacing many outdated references with more current studies. As a result of this revision, the percentage of citations published before 2000 has been reduced from 15% to 2%. We believe these updates offer a more up-to-date synthesis while preserving the integrity of the theoretical framework we propose.

Reviewer #2

I always enjoy reading papers that address big conceptual issues. In this paper, the authors feature a family life course stage that came into the family life course lexicon in western countries in the mid 20th century. I agreed to review the paper because I was intrigued about how someone might now be using the idea of 'empty nest', at a point in history in which there is such immense diversity in family structure that the author rightly identifies in the first section of the paper.

1. *Part of the argument for focusing on the empty nest is that it has become more common in both Western and Asian societies. It would be helpful if this claim could be substantiated. The empty nest period is described as “the postparental phase in parents’ lives when their children have left home and they are no longer engaged in childrearing”. We might be able to determine how many people of a particular age cohort have ever had children, although it could be difficult given that the age range implied in this review could include anyone from age 40 up to very late life. I wonder too if it matters whether ‘their children’ have always lived with them in the same household, or whether they have come together in a blended family, or when the nest is counted as empty. Barbara Mitchell, whose work which is cited in the paper, talks about children who leave and then return and about how in the context of Canada and a housing crisis, children are staying home longer.*

Response:

We thank the Reviewer for the positive feedback and all the constructive comments, which we have incorporated in our revision to enhance the quality of our manuscript. We also appreciate the insightful comment regarding the need to substantiate the claim that the empty nest period has become more common in both Western and Asian societies. In response, we have toned down our language in the introduction and explained how prevailing Asian norms have shifted from one that traditionally encouraged extended multigenerational households to more contemporary arrangements where adult children leave the nest for (primarily) job seeking purposes (p. 4).

Additionally, to substantiate the claim that the empty nest period is increasingly common, following the Reviewer’s suggestion, we have incorporated recent demographic

data and studies that document trends in family structures and living arrangements (pp. 3-4). In summary, in Western societies, the phenomenon of the empty nest has been well-documented over the past several decades. Data from countries such as the United States and Canada indicate a steady increase in the number of middle-aged and older adults living without their children. According to data from the U.S. Census Bureau (2020), there was a 10.6% increase in the number of empty nester households from 2010 to 2020. More than a third of householders aged 45 to 64 (37%) were empty nesters, with the number increasing to 44% for householders aged 65 and above. Similarly, in Canada, there was a 5.5% increase in couples living without children from 2016 to 2021, reaching a total of 25.7% of all households (Statistics Canada, 2022).

In Asian societies, the empty nest phenomenon is also becoming more prevalent. Traditionally, many young adults remained in the parental home, creating multigenerational households due to cultural norms such as filial piety; even when they left, they did so at a later age (Kim et al., 2015; Li & Hung, 2019). However, modernization and urbanization have significantly altered family structures in Asia. Young adults are increasingly migrating to urban areas for employment, leaving their parents in rural regions. Data from recent studies conducted in China and South Korea, which show that urban migration of young adults has led to a significant increase in the number of elderly parents living alone, especially in rural areas (Statistics Korea, 2020; Tao et al., 2023). In China, for example, there are approximately 150 million empty nesters, accounting for more than half of the elderly population (Tao et al., 2023). This number is projected to rise significantly, with empty nesters expected to make up 90% of older adult households by 2030 (Chang et al., 2016; Li et al., 2003; Su et al., 2012). Similar patterns are observed in other Asian countries, where declining fertility rates are compounding the trend. Major economies such as Japan, South Korea, India, Malaysia, and Singapore have all reported plummeting fertility rates,

contributing to a growing population of empty nesters (Park, 2020; Radkar, 2020; Rizkiati et al., 2024; Yang et al., 2022; Yong et al., 2015; Zhan & Huang, 2023).

We also appreciate the Reviewer's observation regarding the complexities involved in defining the empty nest period, particularly in relation to factors such as age range, household composition, and blended families. In response, we have expanded our discussion to include a more nuanced definition of the empty nest period, acknowledging that it can extend beyond middle-aged parents to later life stages, especially in societies where children may remain at home longer due to economic or cultural reasons (pp. 19-20 & pp. 23-24). For more details, please refer to our response to the Reviewer's #2 comment.

2. *The authors argue that loneliness is associated with lack of daily contact with children in some cultures. Does it matter whether they are living alone, whether they depend on their children for financial or other support, whether they live in rural or urban areas or whether a parent moves in with a child in late life? The challenge I see is that the variation across all of these demographic and other social factors is immense. I'd love to see some critical thinking about the usefulness of the concept of empty nest given this diversity.*

Response:

We thank the Reviewer for the constructive suggestion. We have since added a new section "Diversity in family structures" (pp. 18-20) to detail some of these important variations in family arrangements in contemporary society based on extant research. We start off by discussing divorced, single-parent, and stepparent families because there is an extensive body of Western literatures on this topic; specifically young adults from divorced, single-parent, stepparent families are more likely to leave their parents' homes (Bayrakdar &

Coulter, 2017; Li & Hung, 2019; Tomaszewski et al., 2017). Given that older adult parents in these family arrangements do not have a partner they can rely on (i.e., divorced and single-parent), they might experience increase feelings of loneliness. Although topic is less discussed in Asia because of traditional marriage conventions (Nozawa, 2020; Yeung & Park, 2016), some recent work suggests that children in Asia from similar family structures may decide to stay longer in the parental home due to cultural expectations surrounding familial obligations (Kim et al., 2015; Li & Hung, 2019). As recommended by the reviewer, we also discussed the phenomenon of adult children returning to the parental home, or the “boomerang” generation (Mitchell, 2017; Olofsson et al., 2020; Nauck & Ren, 2018). Adults who boomerang are less likely to be financially and emotionally stable, which exacerbates the stress experienced by their parents. Newer research also links the boomerang generation to the crowded nest phenomenon, where older adult parents must maintain their childrearing duties to existing younger children while coping with the demands of children that have boomerang back (Cooklin et al., 2012). Boomerang children is not a phenomenon unique to the West, with some research on the topic conducted in Asian countries like China, Taiwan and Thailand (e.g., Chen et al., 2022; Liao & Paweenawat, 2022). Interestingly, it is more socially accepted in Asia (than in the West) for adult children to return to the parental home after they have their own children to save childcare costs (Liao & Paweenawat, 2022).

Under implications (pp. 23-24), we also discussed heterogeneity both between and within Asian cultures. While many discussions (and empirical research) on socially accepted timelines for departures from the parental homes in Asia often anchor on Confucian values, especially filial piety, which are prevalent in East Asian countries like China, Japan, and South Korea, these values do not uniformly define family norms across the entire continent (Maurer-Fazio et al., 2011; Posadas & Vidal-Fernández, 2013). For instance, in Southeast Asia, extended family support and co-residence are common, but the motivations for these

living arrangements and expectations for adult children are rooted more on cultural and religious beliefs (e.g., Islam, Buddhism, and Hinduism), than in East Asia (Yeung et al., 2018). Similarly, South Asian countries such as India, Pakistan, and Bangladesh maintain strong traditions of multigenerational households, where there is often an expectation that the nest never empties: at least one child—often a son and his wife— remains at home to establish an extended household to care for aging parents (Yeung et al., 2018); violations to these cultural expectations can result in stress, and anxiety for parents (Mitchell & Wister, 2015). These patterns are reinforced by religious and cultural norms that vary widely from East Asian Confucian ideals.

We also make mention of how socioeconomic class and how people living in rural versus urban spaces may experience the empty nest differently (Yi & Wang, 2003). It is likely that in more rural areas, families are more likely to maintain and uphold more traditional structures such as extended or multigeneration families, while urban families may increasingly embrace "Westernized" living arrangements, such as forming nuclear households or having young adults move out at an earlier age. Given this complexity, it is essential for researchers to avoid overgeneralizing when examining the empty nest phenomenon in Asia given the rich tapestry of family forms, norms, and transitions that differ not only between countries but within them. As such, we suggest researchers should engage in more nuanced and localized research to cast spotlight on the uniqueness for how empty nest plays out within different Asian cultural contexts. Culturally sensitive interventions also need to address these variations, and fit contemporary cultural norms, the state of economic development and urbanization to better address the empty nest experiences of older parents and their adult children across the diverse societies of Asia.

3. *It would be a good idea for the authors to be a bit more vigilant about statements that rely on literature that sometimes is decades old. One example is a statement by Rick Settersten from 1998 that “In the West, the common consensus is that children, regardless of gender, should leave home between the ages of 18 to 25 (Settersten, 1998). That statement was made 25 years ago by a family theorist in the U.S. I counted almost 50 references in the paper that are 20+ years old. There’s nothing wrong with citing classic studies. I’d like to see them identified as such.*

Response:

We thank the Reviewer’s valuable feedback regarding the use of older literature in our manuscript. We appreciate the suggestion to be more vigilant about older citations, particularly those like Settersten (1998). In response, we have carefully reviewed the references throughout the manuscript and updated most of the older citations with more recent studies, where applicable. As a result of this revision, the percentage of citations published before 2000 has been reduced from 15% to 2%. Additionally, we have ensured that any older references that remain in the manuscript are identified as classic studies, acknowledging their foundational contribution to the field.

Regarding the specific statement attributed to Settersten (1998) about children leaving home between the ages of 18 to 25, we have replaced this with more recent data that reflect current trends in nest-leaving across various contexts (pp. 14-15). We have also provided more nuanced discussions about how the timing of nest-leaving may vary across countries (Billari & Liefbroer, 2010; Tanner & Arnett, 2016; van den Berg et al., 2021), with socio-economic factors such as housing affordability and labor market conditions playing a key role in these variations (Srinivas, 2019; van den Berg et al., 2021).

4. *In conclusion, I think the premise of the paper is very bold. It needs a bit of tweaking and perhaps a narrower focus to provide the insightful conclusions that we'd like to see in a theoretical paper.*

Response:

We thank the Reviewer for the positive feedback and constructive suggestions regarding the need for further refinement and a narrower focus. We have carefully incorporated these revisions to enhance the quality of the manuscript. By narrowing the focus, refining our arguments, updating references, and providing more nuanced discussions, we believe the revised manuscript now better meets the expectations for a theoretical paper. We hope that these changes address the Reviewer's concerns and improve the overall clarity of our arguments.

References

- Bayrakdar, S., & Coulter, R. (2018). Parents, local house prices, and leaving home in Britain. *Population, Space and Place*, 24(2), e2087. <https://doi.org/10.1002/psp.2087>
- Billari, F. C., & Liefbroer, A. C. (2010). Towards a new pattern of transition to adulthood? *Advances in Life Course Research*, 15(2–3), 59–75.
<https://doi.org/10.1016/j.alcr.2010.10.003>
- Chang, Y., Guo, X., Guo, L., Li, Z., Yang, H., Yu, S., ... & Sun, Y. (2016). Comprehensive comparison between empty nest and non-empty nest elderly: A cross-sectional study among rural populations in Northeast China. *International Journal of Environmental Research and Public Health*, 13(9), 857. <https://doi.org/10.3390/ijerph13090857>
- Chen, Y. J., Matsuoka, R. H., & Wang, H. C. (2022). Intergenerational coresidence living arrangements of young adults with their parents in Taiwan: the role of filial piety. *Journal of Urban Management*, 11(4), 437-449.
<https://doi.org/10.1016/j.jum.2022.07.004>
- Cooklin, A. R., Giallo, R., & Rose, N. (2012). Parental fatigue and parenting practices during early childhood: An Australian community survey. *Child: Care, Health and Development*, 38(5), 654-664. <https://doi.org/10.1111/j.1365-2214.2011.01333.x>
- Kim, K., Cheng, Y. P., Zarit, S. H., & Fingerman, K. L. (2015). Relationships between adults and parents in Asia. In S. T. Cheng, I. Chi, H. Fung, L. Li, & J. Woo (Eds.), *Successful aging* (pp. 85–99). Springer. https://doi.org/10.1007/978-94-017-9331-5_7
- Li, D. M., Chen, T. Y., & Li, G. Y. (2003). The problem of mental health in the elderly in empty-nest family. *Chinese Journal of Gerontology*, 23(7), 405–407. [In Chinese].
- Li, W. D., & Hung, C. Y. (2019). Parental support and living arrangements among young adults in Taiwan. *Journal of Housing and the Built Environment*, 34, 219-233.
<https://doi.org/10.1007/s10901-018-9620-7>

- Liao, L., & Paweenawat, S. W. (2022). Alternative boomerang kids, intergenerational co-residence, and maternal labor supply. *Review of Economics of the Household*, 20(2), 609-634. <https://doi.org/10.1007/s11150-020-09524-9>
- Maurer-Fazio, M., Connelly, R., Chen, L., & Tang, L. (2011). Childcare, eldercare, and labor force participation of married women in urban China, 1982–2000. *Journal of human Resources*, 46(2), 261-294. <https://doi.org/10.3368/jhr.46.2.261>
- Mitchell, B. A. (2017). *The boomerang age: Transitions to adulthood in families*. New Brunswick, NJ: Transaction-Aldine Publishers.
- Mitchell, B. A., & Wister, A. V. (2015). Midlife challenge or welcome departure? Cultural and family-related expectations of empty nest transitions. *The International Journal of Aging and Human Development*, 81(4), 260-280. <https://doi.org/10.1177/0091415015622>
- Nauck, B., & Ren, Q. (2018). Coresidence in the transition to adulthood: The case of the United States, Germany, Taiwan, and mainland China. *Chinese Sociological Review*, 50(4), 443-473. <https://doi.org/10.1080/21620555.2018.1522953>
- Nozawa, S. (2020). Similarities and variations in stepfamily dynamics among selected Asian societies. *Journal of family issues*, 41(7), 913-936. <https://doi.org/10.1177/0192513X209177>
- Olofsson, J., Sandow, E., Findlay, A., & Malmberg, G. (2020). Boomerang behaviour and emerging adulthood: Moving back to the parental home and the parental neighbourhood in Sweden. *European Journal of Population*, 36, 919-945. <https://doi.org/10.1007/s10680-020-09557-x>
- Park, E. H. (2020). Ultra-low fertility and policy response in South Korea: Lessons from the case of Japan. *Ageing International*, 45(2), 191-205. <https://doi.org/10.1007/s12126-020-09365-y>

- Posadas, J., & Vidal-Fernandez, M. (2013). Grandparents' childcare and female labor force participation. *IZA Journal of Labor Policy*, 2, 1-20. <https://doi.org/10.1186/2193-9004-2-14>
- Radkar, A. (2020). Indian fertility transition. *Journal of Health Management*, 22(3), 413-423. <https://doi.org/10.1177/0972063420937925>
- Rizkianti, A., Kistiana, S., Fajarningtias, D. N., Hutasoit, E. F., Baskoro, A. A., Maryani, H., ... & Muthmainnah, M. (2024). Understanding the association between family planning and fertility reduction in Southeast Asia: A scoping review. *BMJ Open*, 14(6), e083241. <https://doi.org/10.1136/bmjopen-2023-083241>
- Tanner, J. L., & Arnett, J. J. (2016). The emergence of emerging adulthood: The new life stage between adolescence and young adulthood. In A. Furlong (Ed.), *Routledge handbook of youth and young adulthood* (2nd ed., pp. 50–56). Routledge.
- Tao, T., Jin, G. Z., & Guo, Y. L. (2023). Empty-nest elderly households in China: Trends and patterns. *Population Research*, 47(1), 58–71. [In Chinese].
- Settersten, R. A. (1998). A time to leave home and a time never to return? Age constraints on the living arrangements of young adults. *Social Forces*, 76(4), 1373–1400. <https://doi.org/10.1093/sf/76.4.1373>
- Srinivas, V. (2019). Explaining the increase in young adults living with parents. *Journal of Economic Issues*, 53(4), 1017-1028. <https://doi.org/10.1080/00213624.2019.1664237>
- Statistics Canada. (2022). Families, households and marital status: Key results from the 2021 Census. Government of Canada.
- Su, D., Wu, X. N., Zhang, Y. X., Li, H. P., Wang, W. L., Zhang, J. P., & Zhou, L. S. (2012). Depression and social support between China's rural and urban empty-nest elderly. *Archives of Gerontology and Geriatrics*, 55(3), 564–569. <https://doi.org/10.1016/j.archger.2012.06.006>

- Tomaszewski, W., Smith, J. F., Parsell, C., Tranter, B., Laughland-Booÿ, J., & Skrbiš, Z. (2017). Young, anchored and free? Examining the dynamics of early housing pathways in Australia. *Journal of Youth Studies*, 20(7), 904-926.
<https://doi.org/10.1080/13676261.2016.1273520>
- U.S. Census Bureau. (2020). America's families and living arrangements: 2020. U.S. Department of Commerce.
- Van den Berg, L., Kalmijn, M., & Leopold, T. (2021). Explaining cross-national differences in leaving home. *Population, Space and Place*, 27(8), e2476.
<https://doi.org/10.1002/psp.2476>
- Yang, S., Jiang, Q., & Sánchez-Barricarte, J. J. (2022). China's fertility change: An analysis with multiple measures. *Population Health Metrics*, 20, 12.
<https://doi.org/10.1186/s12963-022-00290-7>
- Yeung, W. J. J., Desai, S., & Jones, G. W. (2018). Families in southeast and South Asia. *Annual Review of Sociology*, 44(1), 469-495. <https://doi.org/10.1146/annurev-soc-073117-041124>
- Yeung, W. J. J., & Park, H. (2016). Growing up in one-parent families in Asia. *Marriage & Family Review*, 52(1-2), 1-14. <https://doi.org/10.1080/01494929.2015.1124478>
- Yi, Z., & Wang, Z. (2003). Dynamics of family and elderly living arrangements in China: New lessons learned from the 2000 census. *China Review*, 95-119.
- Yong, V., Minagawa, Y., & Saito, Y. (2015). Policy and program measures for successful aging in Japan. In S.-T. Cheng, I. Chi, H. Fung, L. Li, & J. Woo (Eds.), *Successful aging* (pp. 123–146). Springer, Dordrecht. https://doi.org/10.1007/978-94-017-9331-5_6

Zhan, S., & Huang, L. (2023). State familism in action: Aging policy and intergenerational support in Singapore. *China Population and Development Studies*, 7(1), 111–129.

<https://doi.org/10.1007/s42379-023-00132-5>